

# Predicting defects in imbalanced data using resampling methods: an empirical investigation

Ruchika Malhotra[1] and Juhi Jain[2]

[1] Department of Software Engineering, Delhi Technological University (former Delhi College of Engineering), Shahbad Daulatpur, Delhi, India
[2] Department of Computer Science and Engineering, Delhi Technological University (former Delhi College of Engineering), Shahbad Daulatpur, Delhi, India

## ABSTRACT

The development of correct and effective software defect prediction (SDP) models is one of the utmost needs of the software industry. Statistics of many defect-related open-source data sets depict the class imbalance problem in object-oriented projects. Models trained on imbalanced data leads to inaccurate future predictions owing to biased learning and ineffective defect prediction. In addition to this large number of software metrics degrades the model performance. This study aims at (1) identification of useful metrics in the software using correlation feature selection, (2) extensive comparative analysis of 10 resampling methods to generate effective machine learning models for imbalanced data, (3) inclusion of stable performance evaluators—AUC, GMean, and Balance and (4) integration of statistical validation of results. The impact of 10 resampling methods is analyzed on selected features of 12 object-oriented Apache datasets using 15 machine learning techniques. The performances of developed models are analyzed using AUC, GMean, Balance, and sensitivity. Statistical results advocate the use of resampling methods to improve SDP. Random oversampling portrays the best predictive capability of developed defect prediction models. The study provides a guideline for identifying metrics that are influential for SDP. The performances of oversampling methods are superior to undersampling methods.

## INTRODUCTION

Software Defect Prediction (SDP) deals with uncovering the probable future defects. Efficient defect prediction helps in the timely identification of areas in software that can lead to defects in software owing to better resource utilization (*Malhotra, 2016*). Source code metrics give useful insights to software quality attributes like cohesion, coupling, size, inheritance, etc, and are extensively used in developing software defect models (*Basili, Briand & Melo, 1996*; *Singh, Kaur & Malhotra, 2010*; *Radjenović et al., 2013*). Effective models can be generated using the object-oriented (OO) metrics. With the increasing

Corresponding author
Juhi Jain,
juhijain_phdco2k16@dtu.ac.in

complexity of software, early prediction of defects, and assurance of good software quality of projects become difficult tasks. To achieve this task of SDP, machine learning (ML) techniques have been used by several researchers from the past two decades. But software defect data is mostly imbalanced (*Khoshgoftaar, Gao & Seliya, 2010*). This issue of SDP has recently gained a high interest in researchers in the software engineering community. Data is imbalanced if the number of minority classes, in our case, defective classes, is much lower than the majority class, i.e., non-defective classes. This imbalanced distribution of data misguides classifiers while learning the defective class correctly and hence results in biased and inaccurate results. A good defect prediction model will be the one that is trained on similar distribution of instances of defective and non-defective classes. From this point onwards, we will refer to the imbalanced data problem as the class imbalance problem (CIP) because we are dealing with an imbalanced ratio of defective and non-defective classes of software.

CIP is very prominent in many real-life problems like medical diagnosis (*Mazurowski et al., 2008*; *Khalilia, Chakraborty & Popescu, 2011*), fraud detection (*Phua, Alahakoon & Lee, 2004*; *Vasu & Ravi, 2011*), text categorization (*Zheng, Wu & Srihari, 2004*; *Moreo, Esuli & Sebastiani, 2016*), sentiment analysis (*Lane, Clarke & Hender, 2012*), churn prediction (*Burez & Van den Poel, 2009*), *etc*. Researchers are actively participating in finding solutions for this problem in the software engineering domain for the last two decades. One of the prominent solutions is using resampling methods comprising of undersampling and oversampling methods (*Kotsiantis, Kanellopoulos & Pintelas, 2006*). A large number of software metrics adds to the problem of constructing good SDP models. This curse of dimensionality is handled by selecting important and distinct features using correlation feature selection (CFS). CFS is selected as it is a widely accepted feature selection (FS) technique and it has emerged as an effective FS technique in a benchmark study (*Ghotra, McIntosh & Hassan, 2017*). This study deals with the application of resampling methods to handle CIP for 12 Apache datasets and performing a comparative analysis of various SDP models developed using ML techniques. In this study, six oversampling techniques namely Synthetic Minority Over-sampling Technique (SMT), Safe Level Synthetic Minority Over-sampling Technique (SLSMT), Selective Preprocessing of Imbalanced Data (SPD), ADAptive SYNthetic Sampling (ADSYN), Random OverSampling (ROS), Aglomerative Hierarchical Clustering (AHC) and four undersampling techniques namely Condensed Nearest Neighbor + Tomek's modification of Condensed Nearest Neighbor (CNNTL), Random Under-Sampling (RUS), Neighborhood Cleaning Rule (NCL), One Sided Selection (OSS) are explored on different datasets to deal with their imbalanced nature.

The main objective of this paper is to ascertain the importance of dealing with CIP using (1) resampling methods and (2) stable performance evaluators to build correct and effective SDP models. The study also identifies the important internal quality attributes (IQAs) of software on which developers need to focus.

Research Questions (RQs) to achieve the aforementioned objectives are designed as follows.

### RQ1: Which features are repeatedly selected by CFS in software engineering datasets?

RQ1 finds the most important internal metrics that impact the possibility of defect(s) in the software. Software metrics are recognized that software designers and developers should focus on while building the software. We found that defects in software engineering datasets are greatly impacted by LCOM, Ca, Ce, and RFC. LCOM is a cohesion metric whereas Ca, Ce, and RFC are coupling metrics. In any software, high cohesion and low coupling are desired. Therefore, as LCOM, Ca, Ce and RFC play a crucial role in SDP, their values should be monitored while designing the software. This would result in software with fewer defects.

### RQ2: What is the performance of ML techniques on imbalanced data while building SDP models?

RQ2 summarizes the performance of 12 imbalanced datasets with 15 ML techniques in terms of Sensitivity, GMean, Balance, and AUC.

### RQ3a: What is the comparative performance of various SDP models developed using resampling methods?

### RQ3b: Is there any improvement in the performance of SDP models developed using resampling methods?

RQ3 presents the performance of SDP models that are built on balanced datasets. The ratio of defective and non-defective classes is equalized with help of different resampling methods and their performance is compared with their original versions, i.e. when no resampling of classes is done. We empirically proved that all resampling methods improve the performance of SDP models as compared to the models that are trained on original data. The values of AUC, Balance, GMean, and Sensitivity increased with models trained on resampled data, and, hence, these models depict better defect prediction capabilities.

### RQ4: Which resampling method outperforms the addressed undersampling and oversampling methods for building an efficient SDP model?

In RQ3 it is proved that resampling methods produce better SDP models but out of the wide array of resampling methods which resampling method should the developer or researcher choose? RQ4 is generated to find out the resampling method(s) that outperforms the other resampling methods. The results of the Friedman Test followed by the post-hoc Nemenyi Test obtained on AUC, Balance, GMean, and Sensitivity values show that ML models based on ROS and AHC statistically outperforms other models.

### RQ5: Which ML technique performs the best for SDP in imbalanced data?

This study uses 15 ML techniques from different categories. RQ5 tries to explore if there is a statistical difference in the performances of different ML techniques. We constricted the comparison to the models developed using ROS as it was statistically better than other resampling methods except for AHC. Performance values of ROS-based models were comparable but greater than the performance values of AHC-based models. We concluded that statistically ensemble methods and nearest neighbor methods performed better than neural networks and statistical techniques.

The answers to these questions are explored by building ML models on CFS selected features using ten-fold cross-validation. Predictive performances of developed models are evaluated using stable performance evaluators like Sensitivity, GMean, Balance, and AUC. *Kitchenham et al. (2017)* reported that in many studies results are biased because they lack statistical validation. They recommend using a robust statistical test to examine if performance differences are significant or not. Statistical validation is carried out using the Friedman test followed by post hoc analysis that is performed using the Nemenyi test. The conducted study will acquaint developers with useful resampling methods and performance evaluators that will assist them to solve CIP. This study also guides developers and software practitioners about the important metrics that affect the SDP potential of ML models. The result examination ascertained ROS-based and AHC-based ML models as the best defect predictors for datasets related to the software engineering domain. With ROS as a resampling method, nearest neighbors and ensembles exhibited comparable performance in SDP. These models were statistically better than other ML models.

The rest of the paper is organized as follows. "Related work" deals with research work done in SDP for imbalanced data. Empirical Study Design is explained in "Materials and Methods". "Results" expounds on the empirical findings and provides answers to set RQs. Next, "Discussions" summarizes the results and provides the comparison of this study with related studies. "Validity Threats" uncovers the validity threats of this study. Finally, "Conclusions" presents the concluding remarks with potential future directions.

## RELATED WORK

This section presents the related work done in the field of feature selection and resampling solutions proposed in the SDP field.

### Feature selection in SDP

Apache datasets have twenty OO metrics and models developed using all these metrics can hamper the defect prediction capabilities of ML models. The reason is the presence of redundant or irrelevant metrics. This curse of dimensionality can be reduced by using feature reduction strategies. This involves either feature selection—reducing the number of features, or feature extraction—extracting new features from existing ones. This study focuses on feature selection by using a widely acceptable CFS technique. *Ghotra, McIntosh & Hassan (2017)* explored 30 feature selection techniques and concluded CFS as the best feature predictor. They used NASA datasets and PROMISE datasets with 21 ML

techniques. *Balogun et al. (2019)* explored feature selection and feature reduction methods for five NASA datasets over four ML techniques and experimentally concluded that FS techniques did not show consistent behavior for the datasets or ML techniques. Recent studies (*Arar & Ayan, 2017*; *Lingden et al., 2019*) have emphasized the importance of feature selection and the impact of CFS in building efficient models with reduced complexity and computation time. *Balogun et al. (2020)* empirically investigated the effect of 46 FS methods over 25 datasets from different sources using Naïve Bayes and decision trees. Based on accuracy and AUC performance, they concluded CFS was the best performer in the FSS category. Therefore, in this study, CFS is used to reveal the most relevant features.

## Solutions proposed for CIP in SDP

Only 20% of software classes are accountable for the defects in software (*Koru & Tian, 2005*). This principle is enough to explain the reason for the uneven distribution of minority (defective) classes and majority (non-defective) classes. Areas of SDP and software change prediction (*Malhotra & Khanna, 2017*; *Tan et al., 2015*) are explored to handle CIP resulting in promising outcomes. Now, to solve this class imbalance issue, a variety of resampling methods have been proposed in the literature, among which oversampling and undersampling techniques are most widely used. *Liu, An & Huang (2006)* used a combination of oversampling and undersampling techniques for predicting software defects using a support vector machine. The performance of developed models was evaluated using F-Measure, GMean, and ROC curve. Experimentation by *Pelayo & Dick (2007)* showed improvement in GMean values for SDP when the oversampling technique SMT is used with a C4.5 decision tree classifier. *Kamei et al. (2007)* evaluated the effect of ROS, RUS, SMT, and OSS on industrial software. The experimental analysis proved that resampling methods improved the performance of LDA and LR models in terms of F1-measure. *Khoshgoftaar & Gao (2009)* used RUS to handle CIP and also used a wrapper-based feature selection technique for attribute selection. They investigated four different scenarios of sampling techniques and feature selection combinations to evaluate which model has better predictive capability in terms of accuracy and AUC.

*Galar et al. (2011)* performed SDP for imbalanced data using bagging—and boosting—based ensemble techniques with C4.5 as the base classifier. The performance was evaluated using the AUC measure. Some other studies support the application of resampling methods for handling CIP (*Riquelme et al., 2008*; *Pelayo & Dick, 2012*; *Seiffert et al., 2014*; *Rodriguez et al., 2014*). *Shatnawi (2012)* also performed an empirical comparison of defect prediction models built using oversampling techniques with three different classifiers on the eclipse dataset. *Wang & Yao (2013)* investigated ML models built using five different resampling methods on 10 PROMISE datasets and their findings confirmed that models based on resampled data result in better SDP. Experimentations concluded the effective model development with AdaboostNC ensemble. *Wang & Yao (2013)* were not able to conclude on which resampling method should be selected by the software practitioners. We solved this issue by providing detailed statistical analysis

and experimentally proved ROS and AHC to be the better options for researchers and other practitioners.

*Jindaluang, Chouvatut & Kantabutra (2014)* proposed the undersampling technique with the k-centers clustering algorithm which proves to be effective in terms of Sensitivity and F-measure, but they didn't use any stable metric for imbalanced data like GMean or AUC.

*Bennin et al. (2016)* concluded that ROS outperformed the SMT approach when defective classes are less than 20% in the software. This study also empirically proved that ROS is statistically better than SMT, hence supports the conclusions of *Bennin et al. (2016)*.

*Tantithamthavorn, Hassan & Matsumoto (2018)* included a large number of datasets but provided a comparative study with only four resampling methods. They also optimized SMT and found its performance comparable with RUS. We created 1980 models with 12 datasets as compared to their 4242 models with 101 datasets. In this study, we covered 10 resampling methods and provided the breadth of resampling methods. We compared models with 15 ML techniques whereas *Tantithamthavorn, Hassan & Matsumoto (2018)* used only 7 ML techniques. The results of our study are different from those of *Tantithamthavorn, Hassan & Matsumoto (2018)*. The main reasons are that the model evaluation is highly dependent on the nature of the data and the behavior of ML techniques. This problem amplifies by adding the third dimension of resampling methods. Furthermore, *Bennin et al. (2018)* empirically concluded that AUC performance is not improved by the SMT and this contradicts with the results of *Tantithamthavorn, Hassan & Matsumoto (2018)*.

*Agrawal & Menzies (2018)* also proposed the modified SMT by tuning the SMT parameters. They emphasized on the fact that preprocessing, i.e., resampling technique is more important factor than the ML technique used for the construction of prediction model. If data could be better (less skewed), then results would be more reliable. *Malhotra & Kamal (2019)* inspected the impact of oversampling techniques on ML models built with 12 NASA datasets. They demonstrated the improvement in ML models with oversampling and proposed a new resampling method—SPIDER3.

Though many studies have been conducted, still there is no particular set of resampling methods that can be considered the winner of all. These techniques certainly need more replicated studies with different classifiers and different datasets. More often NASA datasets are exploited by researchers for investigating CIP. We have comparatively used Apache datasets to visualize the effect of sampling techniques. Repetitive studies are required to be performed in the future for a fair comparison. Apart from decision tree-based and ensemble-based classifiers, this study used rule-based, neural network-based, and statistical-based machine learners for assessing the predictive capability of models.

## MATERIALS & METHODS

This section describes the components involved in this empirical study. This section describes the framework established to build a classification model for defect prediction

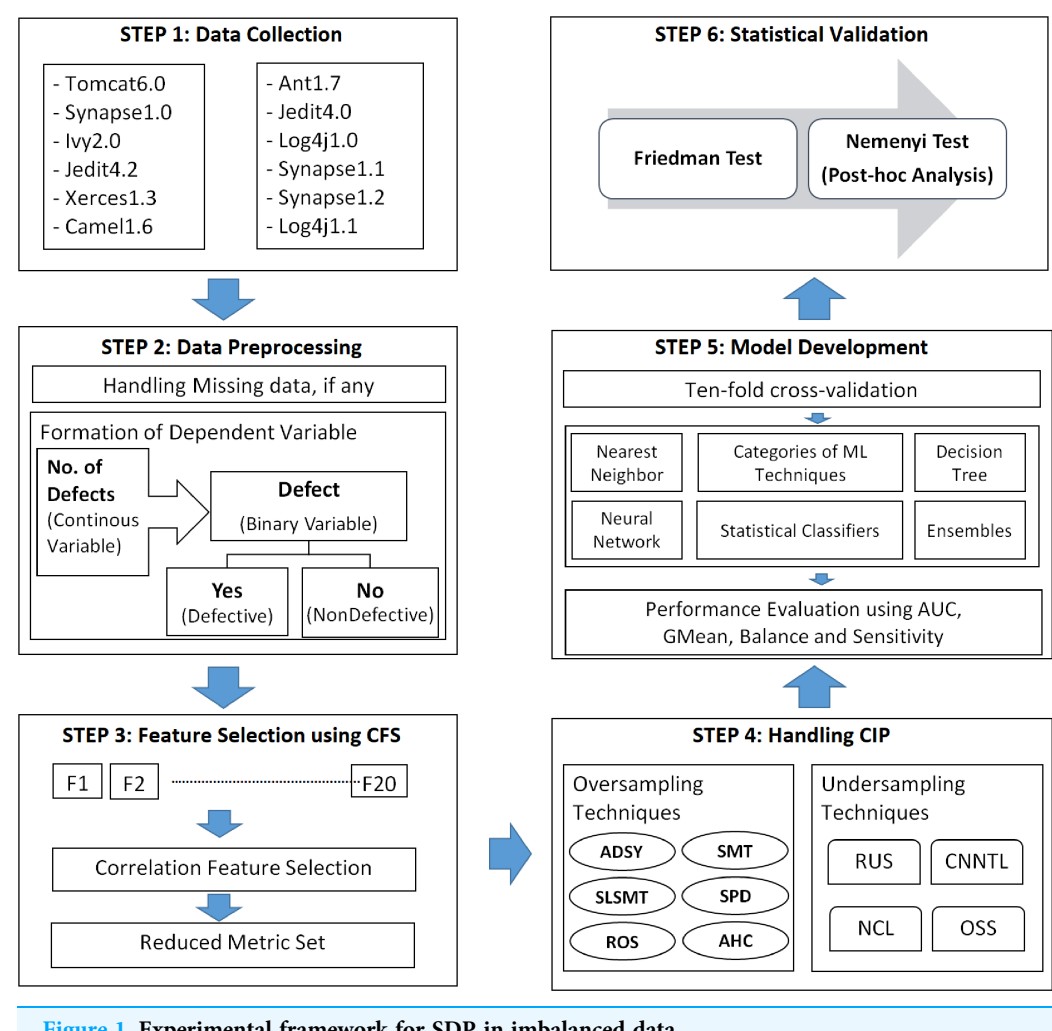

**Figure 1** Experimental framework for SDP in imbalanced data.

from dataset collection to model validation. Figure 1 explains the experimental setup for the study.

## Step1: dataset collection

Datasets mined from the PROMISE repository were used for empirical predictive modeling and validation. These datasets were collected by *Jureczko & Spinellis (2010)*. *Jureczko & Madeyski (2010)* used these datasets to find clusters of similar software projects. The datasets consisted of 20 software metrics and a dependent variable indicating the number of defects in a particular class. The selection of projects was based on the percentage of data imbalance in them. Details of the datasets are provided in Table 1. #Classes denotes the total number of classes in the project, #DClasses denotes the number of defective classes, and #%ageDefects represents the percentage of defective classes in a project. The percentage of defective classes in addressed projects varies from 9.85% to 33.9%. This low percentage represents the imbalanced ratio of defective and non-defective classes in the datasets.

**Table 1 Dataset details.** #Classes denotes the total number of classes in the project, #DClasses denotes the number of defective classes, and #%ageDefects represents the percentage of defective classes in a project.

| Dataset | #Classes | #DClasses | #%ageDefects |
|---|---|---|---|
| Tomcat6.0 | 858 | 77 | 9.85 |
| Synapse1.0 | 157 | 16 | 10.19 |
| Ivy2.0 | 352 | 40 | 11.4 |
| Jedit4.2 | 367 | 48 | 13.1 |
| Xerces1.3 | 453 | 69 | 15.2 |
| Camel1.6 | 965 | 188 | 19.48 |
| Ant1.7 | 745 | 166 | 22.3 |
| Jedit4.0 | 306 | 75 | 24.5 |
| Log4j1.0 | 135 | 34 | 25.2 |
| Synapse1.1 | 222 | 60 | 27 |
| Synapse1.2 | 256 | 86 | 33.6 |
| Log4j1.1 | 109 | 37 | 33.9 |

## Step 2: data preprocessing and identification of variables

The independent variables used in this study are different OO metrics characterizing a software system from various aspects. The OO metrics used in the study include the following metrics:

*Chidamber and Kemerer (CK) Metric suite* (*Chidamber & Kemerer, 1994*): Six popularly used metrics, namely Weighted Methods of a Class (WMC), Depth of Inheritance Tree (DIT), Number of Children (NOC), Lack of Cohesion in Methods (LCOM), Response For a Class (RFC) and Coupling Between Objects (CBO), are incorporated in this metric suite. This metric suite has been validated in many empirical studies for developing SDP models.

*Quality Model for Object-Oriented Design metric suite (QMOOD)* (*Bansiya & Davis, 2002*): The study uses few metrics from this metric suite namely Number of Public Methods (NPM), Data Access Metric (DAM), Method of Functional Abstraction (MFA), Measure of Aggression (MOA) and Cohesion among Methods of a class (CAM) along with CK metrics for developing defect prediction models. These metrics are also well exploited in related studies to develop effective software quality prediction models.

*Other metrics*: Few other metrics have also been used in this study as independent variables in addition to the above metrics that are widely used by researchers. These metrics are—Efferent Coupling (Ce), Afferent Coupling (Ca), Lines of Code (LOC), Coupling Between Methods of a Class (CBM), Average Method Complexity (AMC), and the variant of LCOM (LCOM3). Ce and Ca are proposed by *Martin (1994)* and LOC, CBM, AMC, and LCOM3 are proposed by *Henderson-Sellers (1995)*. Inheritance Coupling (IC), maximum Cyclomatic complexity (Max_cc), and average cyclomatic complexity (Avg_cc) are also used in addition to the above metrics.

Details of metrics used can be referred to from http://gromit.iiar.pwr.wroc.pl/p_inf/ckjm/metric.html.

| Table 2 | OO features selected by CFS. |
|---------|------------------------------|
| **Dataset** | **Features selected by CFS** |
| Tomcat6.0 | CBO, RFC, LCOM, LOC, MOA, AMC, Max_CC, Avg_CC |
| Synapse1.0 | RFC, LCOM, Ce, LOC, DAM, MFA, CAM, IC, AMC, Max_CC, Avg_CC |
| Ivy2.0 | RFC, Ce, LOC, MOA, AMC, NPM, CBO, WMC, LCOM, LCOM3, CAM |
| Jedit4.2 | CBO, RFC, Ca, Ce, NPM, LCOM3, MOA, LOC, CAM, CBM, LCOM, AMC, Max_CC |
| Xerces1.3 | WMC, Ce, LCOM, MOA, DAM, IC, CBM, AMC |
| Camel1.6 | DIT, NOC, CBO, LCOM, Ca, NPM, LCOM3, CAM, IC, CBM, AMC, Max_CC, Avg_CC |
| Ant1.7 | CBO, RFC, LCOM, Ce, LOC, MOA, CAM, AMC, Max_CC |
| Jedit4.0 | WMC, DIT, CBO, RFC, LCOM, Ca, Ce, NPM, LCOM3, LOC, DAM, MOA, CBM, Max_CC, Avg_CC |
| Log4j1.0 | WMC, CBO, RFC, LCOM, Ce, Ca, NPM, LOC, DAM, CAM, Avg_CC |
| Synapse1.1 | DIT, CBO, RFC, LCOM, Ce, DAM, MFA, CAM, Max_CC |
| Synapse1.2 | WMC, CBO, RFC, Ca, Ce, LOC, MOA, CBM, CAM, AMC, Max_CC |
| Log4j1.1 | WMC, RFC, LCOM, Ce, NPM, LCOM3, MOA, MFA |

Datasets are checked for any inconsistencies like missing data or redundant data. If there are data inconsistencies, the model prediction would be biased. Therefore, it was important to clean the data before using it for model development. Datasets collected contain a continuous variable representing the number of defects. It was converted into a binary variable by replacing '0' with 'No' and natural numbers with 'yes'. The binary dependent variable, 'defect' with two possible values are 'yes' and 'no' reflects whether the software class is defective or not.

## Step 3: feature selection

All these metrics may not be important in the concern of predicting defects in the early stages of project development. This study employs CFS for the identification of significant metrics. A review by *Malhotra (2015)* has revealed that CFS is the most commonly used feature selection technique. CFS is used in this study for selecting features because it is the most preferred FS technique in SDP literature (*Ghotra, McIntosh & Hassan, 2017*; *Arar & Ayan, 2017*; *Lingden et al., 2019*; *Balogun et al., 2020*). CFS performs the ranking of features based on information gain and identifies the optimal subset of features. The features in the subset are highly correlated to the class label 'defect' and are uncorrelated or less correlated with each other. CFS is incorporated to minimize the multicollinearity effect.

A list of metrics selected by CFS for each dataset is presented in Table 2.

## Step 4: handling CIP

This step involves applying resampling methods for treating CIP. These methods are implemented using a knowledge extraction tool based on evolutionary learning (KEEL) (*Alcalá-Fdez et al., 2011*). Six oversampling methods and four undersampling methods were applied to create a balance between majority and minority classes. Details of these resampling methods are given in Table 3.

**Table 3 Resampling methods used in the study.**

| Resampling Methods | | Description |
|---|---|---|
| **Oversampling Methods** | ADAptive SYNthetic Sampling (ADSYN) (*He et al., 2008*) | In ADASYN, synthetic samples are generated by finding the density distribution of minority classes. Density distribution is computed using a k-nearest neighbor with Euclidian distance. It is an extension of the Synthetic Minority Oversampling Technique. It focuses on the samples that are hard to classify. |
| | Synthetic Minority Over-sampling Technique (SMT) (*Chawla et al., 2002*) | The number of minority class samples is increased by generating artificial samples in direction of k nearest neighbors of minority class samples. If one neighbor is selected, then one synthetic sample is generated corresponding to that original minority sample resulting in 100% oversampling of minority classes. In this study, k = 5 results in 500% oversampling of minority classes. |
| | Safe Level Synthetic Minority Oversampling Technique (SLSMT) (*Bunkhumpornpat, Sinapiromsaran & Lursinsap, 2009*) | Unlike the SMT version, where synthetic samples are generated randomly, in SLSMT first safe levels are calculated that helps in determining safe positions for generating the synthetic samples. If the safe level value is close to 0, it is considered noise and if the safe level value is close to k for k nearest neighbor implementation, then it is considered safe resulting in producing synthetic minority samples there. |
| | Selective Preprocessing of Imbalanced Data (SPD) (*Stefanowski & Wilk, 2008*) | This technique categorizes the sample as either safe or noise based on the nearest neighbor rule where distance measurement is done using a heterogeneous value distance metric. If an instance is accurately classified by its k nearest neighbors, it is considered safe otherwise it is considered noise and then discarded. |
| | Random OverSampling (ROS) (*Batista, Prati & Monard, 2004*) | ROS is a very simple oversampling technique in which minority class instances are replicated at random with the sole aim of creating a balance between majority and minority class instances. |
| | Agglomerative Hierarchical Clustering (AHC) (*Cohen et al., 2006*) | In AHC, each class is decomposed into sub-clusters and synthetic samples are generated corresponding to cluster prototypes. Since artificial samples are created as centroids of sub-clusters of classes, they, therefore extract the characteristics of that class and represent better samples than randomly generated samples. |
| **Undersampling Methods** | Random UnderSampling (RUS) (*Batista, Prati & Monard, 2004*) | RUS, like ROS, is a non-heuristic technique. But in this instead of replicating minority class instances, majority class instances are removed with aim of creating a balance between majority and minority class instances. The problem with this technique is that some important or useful data may be rejected as it is based on random selection. |
| | Condensed Nearest Neighbor(CNN) + Tomek's modification of Condensed Nearest Neighbor (CNNTL) (*Cohen et al., 2006*) | First CNN is applied to find a consistent subset of samples that helps to eliminate majority class samples that are far from the decision border. Then Tomek links (*Tomek, 1976*) are made between samples. If it exists between any two samples, then either both are borderline samples or one of them is noise. Samples with Tomek links that fit in majority classes are removed. |
| | Neighborhood Cleaning Rule (NCL) (*Laurikkala, 2001*) | NCL uses the edited nearest-neighbor (ENN) rule to remove majority class samples. For each training sample Si, first it finds three nearest neighbors. If Si= majority class sample, then discard it if more than two nearest neighbors incorrectly classifies it. If Si = minority class sample, then discard the nearest neighbors if they incorrectly classify it. |
| | One Sided Selection (OSS) (*Kubat & Matwin, 1997*) | OSS and CNNTL have similar working. The difference lies in the order of the application of CNN and the determination of Tomek links. OSS identifies unsafe samples using Tomek links and then applying condensed nearest neighbor (CNN). Noisy and borderline samples are considered unsafe. Small noise may result in the flipping of the decision border of the borderline samples; therefore, they are also considered unsafe. CNN eliminates the majority of samples that are far away from decision boundaries. |

| Table 4 Confusion matrix. | | |
|---|---|---|
| | **Predicted Positive** | **Predicted Negative** |
| **Actual Positive** | True Positive (TP) | False Negative (FN) |
| **Actual Negative** | False Positive (FP) | True Negative (TN) |

## Step 5: performance evaluation and model development

### Performance evaluators

When it comes to imbalanced data, the selection of appropriate performance evaluators plays a critical role. These measures can be calculated by using the confusion matrix shown in Table 4. TP represents the number of defective classes predicted correctly. TN represents the number of non-defective classes predicted correctly. FP represents the number of non-defective classes that are wrongly predicted as defective classes. FN represents the number of defective classes that are wrongly predicted as non-defective classes.

The use of accuracy to evaluate performance is specious when data is imbalanced. Instead, robust performance evaluators like AUC, GMean, and Balance should be used in the class imbalance framework. The Sensitivity indicates the probability of correctly predicted defective classes out of total defective classes. whereas specificity refers to the probability of identifying non-defective classes correctly. Sensitivity or True Positive Rate (TPR) is defined as

$$Sensitivity = \frac{TP}{TP + FN} \tag{1}$$

GMean maintains a balance between both these accuracies (*Li et al., 2012*). Therefore, it is wise to use GMean as an effective measure to assess imbalanced data. GMean is defined as the geometric mean of Sensitivity and specificity for any classifier.

$$GMean = \sqrt{sensitivity * specificity} \tag{2}$$

where

$$Specificity = \frac{TN}{TN + FP} \tag{3}$$

Balance corresponds to the Euclidean distance between a pair of Sensitivity and False Positive Rate (FPR) (*Li et al., 2012*). FPR is the probability of false alarm. It exemplifies the proportion of non-defective classes that are misclassified as defective classes amongst actual non-defective classes. Balance can be computed as—

$$Balance = 1 - \sqrt{\frac{\left(0 \frac{FPR}{100}\right)^2 + \left(1 \frac{Sensitivity}{100}\right)^2}{2}} \tag{4}$$

where

$$\text{FPR} = \frac{FP}{TN + FP} \tag{5}$$

The area under the curve (AUC) is widely accepted as a consistent and robust performance evaluator for predictions in imbalanced data (*Fawcett, 2006*; *Lessmann et al., 2008*; *Malhotra & Khanna, 2017*). It is threshold independent and can handle skewed data. It is a measure to distinguish between the two classes. The range of AUC is (0, 1). Higher the AUC value the better the prediction model. AUC value of 0.5 signifies that the model cannot differentiate between the two classes. AUC values from 0.7 to 0.8 are considered acceptable. AUC values greater than 0.8 are considered excellent.

### Model development using ML techniques

This study developed models based on 15 ML techniques. Ten-fold within project cross-validation is carried out to reduce the partitioning bias. Data was divided into ten partitions. Nine partitions were used for the training part and the remaining one partition was used for the testing part. Then performance evaluators are averaged across ten folds. The ML parameters that were used in experiments created in this study are noted in Table 5. ML techniques used in this study can be divided into five major categories as described below.

*Statistical techniques*

- *Naive Bayes (NB)* (*John & Langley, 1995*): Naïve Bayes is a probability-based classifier that works on the Bayes theorem. It is an instance-based learner that computes class wise conditional probabilities. Features need to be conditionally independent with each other. It provides fair results even in violation of this assumption. This ML technique works well for both categorical and numerical variables.
- *Logistic Regression (LR)* (*Le Cessie & Van Houwelingen, 1992*): Logistic Regression is also a probabilistic classifier used for dichotomous variables and assumes that the data follows Gaussian distribution. It works well in case of assumption desecration. During training coefficient values are minimized by ridge estimator to solve multicollinearity and this makes the model simpler. The algorithm runs until it converged.
- *Simple Logistic (SL)* (*Sumner, Frank & Hall, 2005*): SimpleLogistic uses LogitBoost to construct logistic regression models. LogitBoost uses the logit transform to predict the probabilities. With each repetition, one simple regression model is added for each class. The process terminates when there is no more reduction in classification error.
- *LogitBoost (LB)* (*Friedman, Hastie & Tibshirani, 2000*): LogitBoost is an additive logistic regression with a decision stump as the base classifier. It maximizes the likelihood and, therefore, generalizes the linear logistic model. The base classifier taken is the decision stump which considers entropy for classification.

*Neural networks*

- *MultiLayerPerceptron (MLP) (*Rojas & Feldman, 2013*):* It is a backpropagation neural network that uses sigmoid function as the activation function. The number of hidden layers in the network is determined by the average of the number of attributes and total classes for a particular dataset. The error is backpropagated in every epoch and reduced via gradient descent. The network is then learned based on revised weights.

*Nearest neighbors*

- *IBk* (*Aha, Kibler & Albert, 1991*): IBk is an instance-based K-nearest neighbor learner. It calculates the Euclidian distance measure of the test sample with all the training samples to find its 'k' nearest neighbors. It then assigns the class label to the testing instance based on the majority classification of nearest neighbors. Only one nearest neighbor is determined with k = 1 and the class label of that nearest neighbor is assigned to the testing instance.
- *Kstar* (*Cleary & Trigg, 1995*): Like Ibk, Kstar is also an instance-based learning algorithm. The difference between the two techniques is about the similarity measures they use. IBk exploits Euclidian distance and Kstar uses the similarity measure based on entropy. Kstar exhibits good classification competence for noisy and imbalanced data.

*Ensembles*

- *AdaboostM1 (ABM1)* (*Quinlan, 1993*; *Freund & Schapire, 1996*): AaboostM1 is an ensemble technique where numbers of weak classifiers are used iteratively to improve the overall performance. It augments the performance of weak learners by adjusting the weak hypothesis returned by the weak learner. The base decision tree used in ABM1 is J48. J48 learns from the previous trees about misclassified instances and calculates the weighted average. NumIterations in parameters represent the number of classifiers involved in this ensemble. This technique helps in reducing the bias in the model.
- *Bagging (Bag)* (*Quinlan, 1993*; *Breiman, 1996*): Bagging or bootstrap aggregation is also one of the ensemble techniques that improve the predictive capability of base classifiers by making bags of training data. Models work in parallel and their results are averaged. Bagging reduces the variance. The number of bags used for experimentation is 10 and the base classifier used is J48.
- *Iterative Classifier Optimizer* (ICO): LogitBoost is used as the iterative classifier in this technique. Cross-validation is utilized for its optimization. In the experiments conducted, it goes through 50 iterations to decide for the best cross-validation.
- *Logistic Model Tree (LMT)* (*Landwehr, Hall & Frank, 2005*): Logistic Model Tree is a meta-learning algorithm that uses logistic regression at leaf nodes for classification. A combination of linear logistic regression and decision tree helps in dealing with the bias-variance tradeoff. This technique is robust to missing values and can handle numeric as well as nominal attributes.

- *Random Tree (RT)* (*Leo, 2001*; *Sumner, Frank & Hall, 2005*): Random Tree is an ensemble-based supervised learner where different trees are constructed from the same population. Random samples of the population are generated to form different trees with a random selection of features. After bags are constructed, models are developed and majority voting is performed to classify the class.
- *Random SubSpace method (RSS)* (*Ho, 1998*): Random SubSpace is used to construct random forests. Randomly feature subsets are selected to generate multiple trees. Bagging is performed with Reptree. Reptree is faster than the basic decision tree and generates multiple trees in each iteration. It then selects the tree whose performance is the best.

*Decision trees*

- *Pruning rule-based classification tree (PART)* (*Frank & Witten, 1998*): PART is a rule-based learning algorithm that exploits partial C4.5 decision trees and generates rules at each iteration. PART stands for a pruning rule-based classification tree. The rule that results in the best classification is selected. MDL is used to find the optimal split. smaller the confidence factor more will be the pruning done.
- *J48* (*Quinlan, 1993*): J48 is a JAVA implementation of the C4.5 decision tree. It follows the greedy technique to build a decision tree and uses the gain ratio as splitting criteria. Leaf nodes are the classification labels-defective and non-defective and rules can be derived by traversing from root to leaf node. It generates a binary tree, and one-third of the data is used for reduced error pruning.

## Step 6: statistical validation

The results need to be statistically verified because without the involvement of statistics results may be misleading (*Frank & Witten, 1998*). *Kitchenham et al. (2017)* emphasized employing robust statistical tests for validating the experimental results. For statistical validation, we can use either parametric or nonparametric tests. The nonparametric Friedman statistical test (*Friedman, 1940*) and the Nemenyi Test are exercised in this study because software data do not follow a normal distribution (*Demšar, 2006*). The Friedman test is executed for different performance evaluators for establishing the statistical difference amongst the performance of developed SDP models. We need to compare several ML models built for several datasets. Therefore, Friedman rankings are computed using the Friedman test. Mean ranks are determined with the help of actual values of performance measure and then these ranks are exploited to perform post-hoc Nemenyi test. If the Friedman test results tend to be positive, post hoc analysis is carried by the Nemenyi test to find pair-wise significant differences. Nemenyi test is executed to determine the technique that statistically outperforms others. The Friedman test and the Nemenyi test are the non-parametric alternatives of the parametric ANOVA test and Tukey test. Both tests are performed using a 95% confidence interval. Hypotheses are set for the corresponding test and we need to accept or refute hypotheses at $\alpha = 0.05$.

**Table 5 Parameters used in ML techniques.**

| Category | ML technique | Parameter settings |
|---|---|---|
| Statistical ML techniques | NB | useKernelEstimator = false, displayModelInOldFormat = false, useSupervisedDiscretization = false |
| | LR | Ridge: 1.0E−8, useConjugateGradientDescent = false, maxIts = −1 |
| | SL | Heuristic Stop = 50, Max Boosting Iterations = 500, useCrossValidation = True, weightTrimBeta = 0.0 |
| | LB | Zmax = 3.0, likelihoodThreshold = −1.7976931348623157E308, numIterations = 10, numThreads = 1, poolSize = 1, seed = 1, shrinkage = 1.0, useResampling = False, weightThreshold = 100 |
| Neural Networks | MLP | Hidden layer = a, Learning rate = 0.3, Momentum = 0.2, Training time = 500, Validation threshold =20 |
| Nearest Neighbour Methods | IBk | KNN = 1, nearestNeighbourSearchAlgorithm = LinearNNSearch, |
| | K* | globalBlend = 20, entropicAutoBlend = False |
| Ensemble Methods | ABM1 | numIterations = 10, weightThreshold = 100, seed =1, classifier = J48: confidenceFactor = 0.25, minNumObj = 2, numFolds = 3, seed = 1, subtreeRaising = True, useMDLcorrection = True |
| | Bag | numIterations = 10, numExecutionSlots = 1, seed =1, bagSizePercent = 100, classifier = J48: confidenceFactor = 0.25, minNumObj = 2, numFolds = 3, seed = 1, subtreeRaising = True, useMDLcorrection = True |
| | ICO | evaluationMetric = RMSE, lookAheadIterations = 50, numFolds = 10, numRuns = 1, numThreads = 1, poolsize = 1, seed = 1, stepSize = 1, iterativeClassifier = LogitBoost |
| | LMT | errorOnProbabilities = False, fastRegression = True, minNumInstances = 15, numBoostingIterations = −1, weightTrimBeta = 0.0 |
| | RT | KValue = 0, breakTiesRandomly = False, maxDepth = 0, minNum = 1, minVarianceProp = 0.001, numFolds = 0, seed = 1 |
| | RSS | numExecutionSlots = 1, numIterations = 10, seed = 1, subSpaceSize = 0.5, classifier = Reptree: initialCount = 0.0, maxDepth = -1, minNum = 2.0, minVarianceProp = 0.001, numFolds = 3, seed = 1 |
| Decision Tree | PART | confidenceFactor = 0.25, minNumObj = 2, numFolds = 3, reducedErrorPruning = False, seed = 1, useMDLcorrection = True |
| | J48 | confidenceFactor = 0.25, minNumObj = 2, numFolds = 3, seed = 1, subtreeRaising = True, useMDLcorrection = True |

# RESULTS

## RQ1: Which features are repeatedly selected by CFS in software engineering datasets?

A total of 20 OO metrics of datasets fundamentally define the IQAs of the software and can be grouped into cohesion, coupling, size, complexity, inheritance, encapsulation, and composition metrics. OO metrics corresponding to each IQA are presented in Table 6. #Selected denotes the number of times a particular metric is selected by CFS for all datasets. LCOM was selected by 11 datasets whereas the Ce metric was selected by 10 datasets. This RQ contemplates the metrics that are important for SDP. The weightage of each metric that was selected by CFS for 12 datasets is considered and their proportion selection was determined for each IQA.

In cohesion metrics, LCOM, CAM, and LCOM3 were chosen by 91.67%, 66.67%, and 41.67% of datasets respectively. The cumulative proportion of the selection of cohesion metrics is 66.7%. This shows that cohesion metrics are important for SDP and while developing the software, developers can focus more on LCOM and CAM values. Similarly, the composition metric (MOA) was selected by 66.67% of the datasets.

**Table 6 Proportion selection of IQAs and CFS selected metrics.**

| IQA | OO Metric | #Selected | Proportion selection |
|---|---|---|---|
| **Cohesion** | LCOM | 11 | 0.667 |
| | CAM | 8 | |
| | LCOM3 | 5 | |
| **Composition** | MOA | 8 | 0.667 |
| **Size** | WMC | 6 | 0.583 |
| | NPM | 6 | |
| | LOC | 8 | |
| | AMC | 8 | |
| **Coupling** | Ca | 9 | 0.556 |
| | Ce | 10 | |
| | CBO | 5 | |
| | RFC | 9 | |
| | CBM | 3 | |
| | IC | 4 | |
| **Complexity** | Avg_CC | 4 | 0.5 |
| | Max_CC | 8 | |
| **Encapsulation** | DAM | 5 | 0.462 |
| **Inheritance** | NOC | 1 | 0.194 |
| | MFA | 3 | |
| | DIT | 3 | |

Exploring Table 6, though the cumulative proportion of coupling metrics is 55.6% its significance can be judged by selecting the top three selected metrics—RFC, Ca, and Ce. RFC is picked by 10 datasets whereas Ca and Ce are opted by 9 datasets each. Considering only these three metrics, the proportion selection of coupling metrics raises from 55.65% to 77.78%.

In size metrics, LOC and AMC are more preferred software metrics for defect prediction. In all datasets, the least selected metrics belong to the inheritance category. Therefore, resource investment can be done wisely by developers. The number of times any metric is selected for all the datasets guides developers and software practitioners in determining its worth for SDP.

## RQ2: What is the performance of ML techniques on imbalanced data while building SDP models?

A box and whisker diagram (boxplot diagram) graphically represents numerical data distributions using five statistics: (a) the smallest observation, (b) lower quartile ($Q_1$), (c) median, (d) upper quartile ($Q_3$), and (e) the largest observation. The box is constructed based on the interquartile range (IQR) from $Q_1$ to $Q_3$. The median is represented by the line inside the box. The whiskers at both ends indicate the smallest observation and the largest observation.

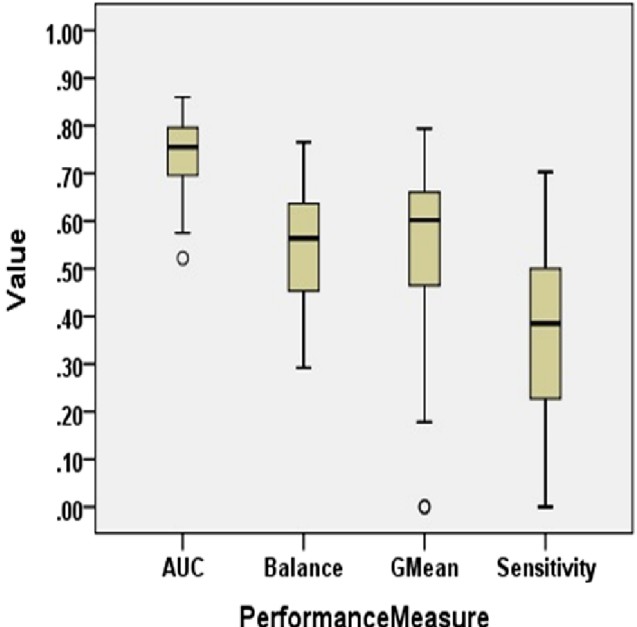

**Figure 2** **Boxplot diagram representing performance measures using original data (NS Case).**

Figure 2 presents the Boxplot diagrams to visually depict the defect predictive capability of ML models on imbalanced data in terms of AUC, Balance, GMean, and Sensitivity.

### AUC analysis
The AUC of ML models of imbalanced data varies from 0.52 to 0.86 with a median value of 0.76. Only 55.6% of models have AUC greater than 0.75. AUC is less than 0.85 in 97.8% of cases. Log4j1.1 depicted the best AUC value of 0.86 with NB. Analysis of results obtained on experimentation reveals that statistical techniques, NB and SL, have performed fairly well in terms of AUC, depicting the highest AUC values for Log4j1.1, Ant1.7, Jedit4.2, Log4j1.0, and Tomcat6.0. Similarly, LMT models were able to predict the highest AUC values for five datasets. Overall, 25% of ML models have AUC less than 0.7.

### Balance analysis
The range of Balance in No Resampling (NS) case is from 29.20 to 76.55. In Tomcat6.0, only 59.01 value was achieved as the maximum value by NB. Balance values achieved by RSS, PART, and LMT are comparatively lower than other ML techniques. The median value for Balance for all models in the NS case is 56.35. Ivy2.0, Synape1.0, and Tomcat6.0 attained the maximum Balance value with NB. IBk also resulted in maximum Balance values for Jedit4.0, Synapse1.1, and Xerces1.3. 68.3% of datasets have a Balance value of less than 65. Only 3.3% of datasets have a Balance value greater than 75.

### GMean analysis
When no resampling method is used, GMean ranges from 0 to 0.79. The median value of GMean for all datasets is observed as 0.6. NB achieved the highest GMean values for

58.33% of datasets. Considering the models for all datasets with 5 different ML techniques, only 14.4% of models achieved GMean greater than 0.7. 30.6% of models have a GMean value less than 0.5.

### Sensitivity analysis

Sensitivity is less than 60% in 93.9% of cases. The median value of Sensitivity is only 0.39. This supports the low predictive capability of developed models when CIP is not handled. Sensitivity values lie between 0 and 0.63. Only 0.6% of cases have a Sensitivity greater than 70% which is not an acceptable achievement for any prediction model. The maximum Sensitivity value obtained in the NS case is 0.7 by IBk, the nearest neighbor technique, in the Log4j1.1 dataset.

Thus, the overall performance of SDP models developed using machine learning algorithms on imbalanced data is not satisfactory for high-quality predictions.

## RQ3a: What is the comparative performance of various SDP models developed using resampling methods?

## RQ3b: Is there any improvement in the performance of SDP models developed using ML techniques on the application of resampling methods?

To answer these questions, we exploited performance evaluators—Sensitivity, GMean, Balance, and AUC values that are calculated with help of a confusion matrix obtained by ten-fold cross-validation-trained models developed using resampling methods. Boxplot diagrams are generated and presented in Fig. 3 to visually depict the defect predictive capability of ML models in terms of AUC, Balance, GMean, and Sensitivity on resampled data. Median values of the best resampling method and NS scenario for AUC, Balance, GMean, and Sensitivity are recorded in Figs. 4, 5, 6, and 7 in form of bar graphs. We have analyzed the performance of developed models based on mean and median values obtained for the cumulative ML techniques of considered performance evaluators. NS cases are included to provide a fair comparison with the resampling-based models.

### Comparative performance of various SDP models developed using resampling methods

For all the datasets, the median values are reported for NS and the resampling method that yields the maximum median performance value in bar graphs. Comparison of boxplot diagram in Fig. 2 with boxplot diagram in Fig. 3 shows the rise in the median line, quartiles, and the highest value achieved by models that were built using resampled data.

### AUC analysis

On the application of resampling methods, 51.4% of models achieved AUC greater than 0.8. From the bars in Fig. 4, it is visible that ROS performance is the best amongst others. ROS attained the highest mean and median value for 75% of the datasets-Ant1.7, Camel1.6, Ivy2.0, Jedit4.0, Jedit4.2, Log4j1.0, Synapse1.0, Tomcat6.0, and Xerces1.3. It also showed the highest mean value for Synapse1.1. Other resampling methods like AHC, SMT,

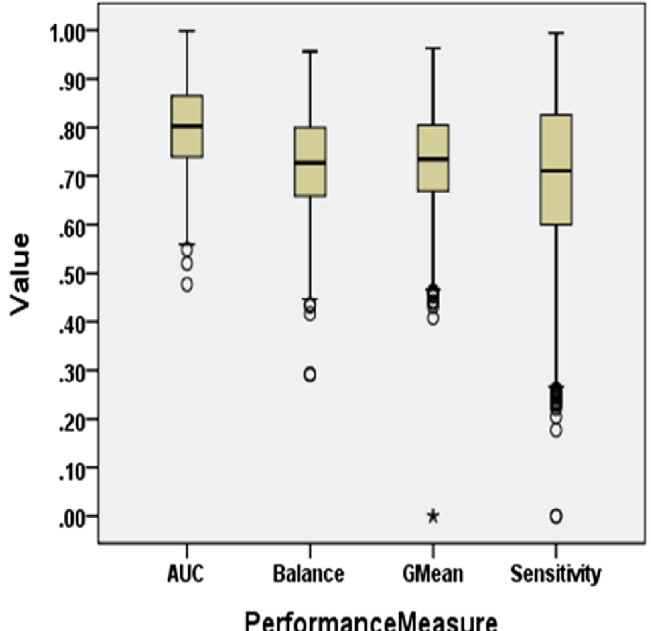

**Figure 3 Boxplot diagram representing performance measures using resampled data.**

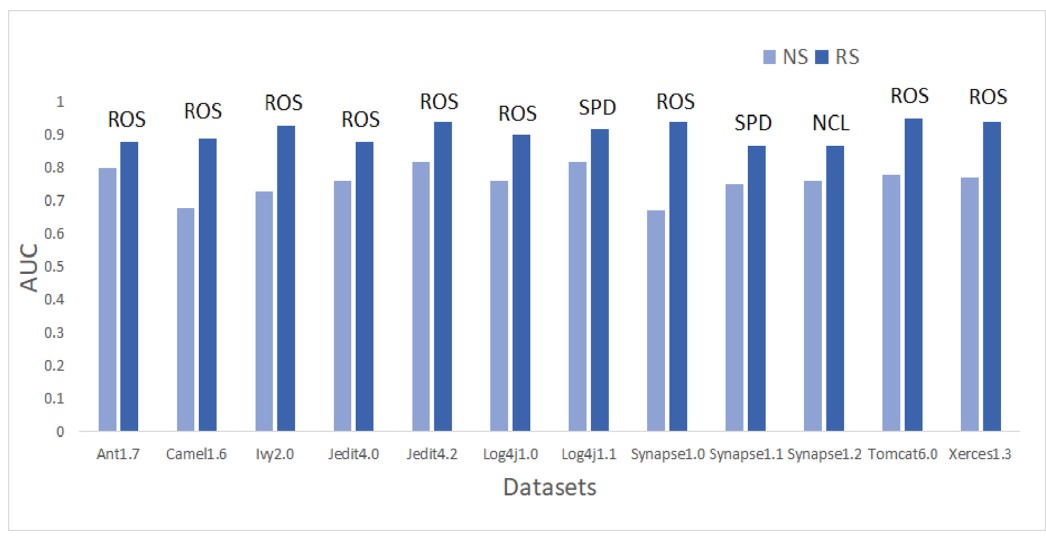

**Figure 4 Dataset-wise comparison of median AUC values of NS and RS.**

and SPD also have depicted good performance in terms of AUC. SPD got the highest median value for Log4j1.1 (0.92) and Synapse1.1 (0.87). In undersampling methods, only NCL results can be considered progressive. NCL was able to manage to secure the highest median value for only one dataset, i.e., Synapse1.2. The highest AUC value of 1 is

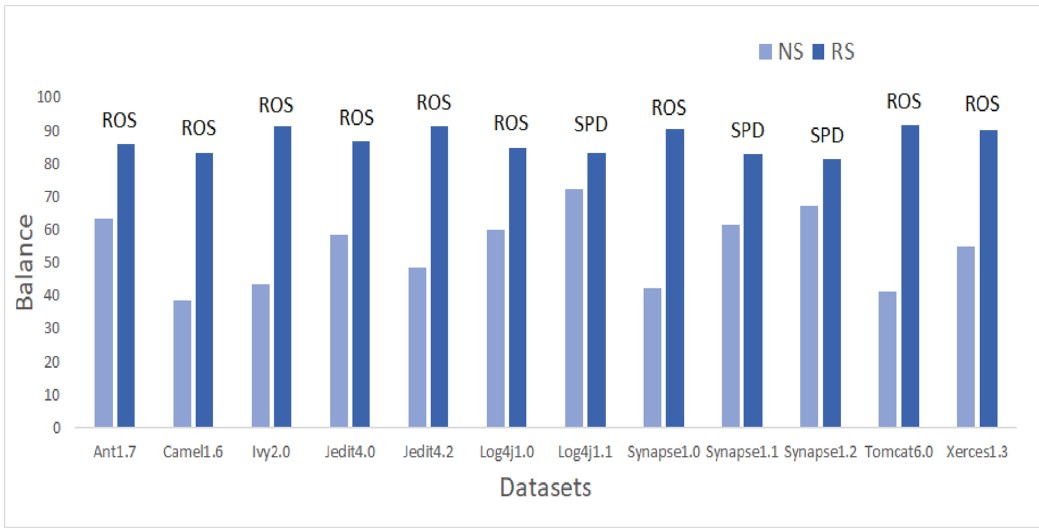

**Figure 5 Dataset-wise comparison of median balance values of NS and RS.**

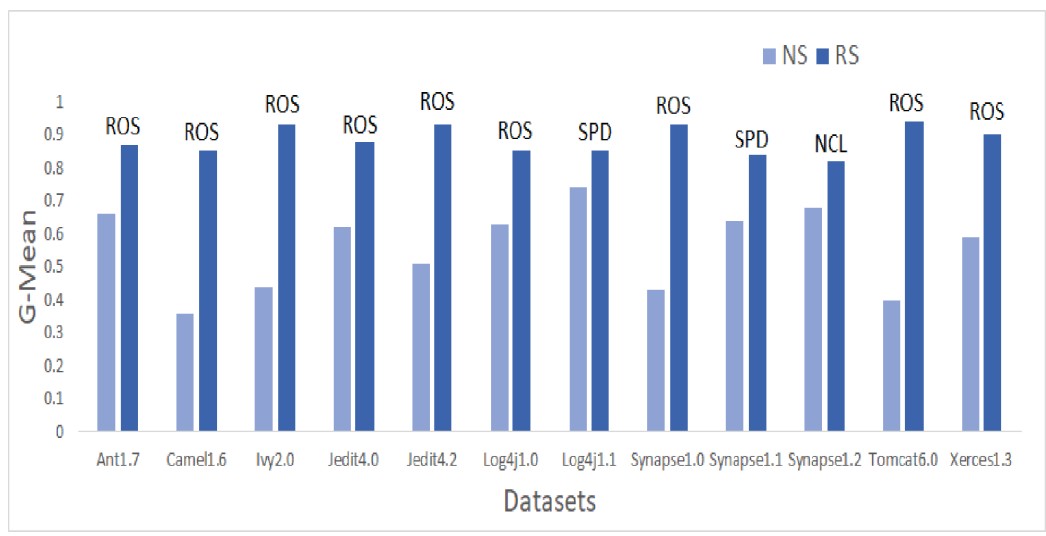

**Figure 6 Dataset-wise comparison of median GMean values of NS and RS.**

achieved by Jedit4.2 and Tomcat6.0 datasets. 13.9% of models have AUC greater than 90%, which is a remarkable improvement.

*Balance analysis*

Referring to Fig. 5, according to the Balance performance evaluator values achieved in predictive modeling, ROS performance seemed to outperform the other resampling methods. ADSYN could achieve a maximum of 86.24 Balance value for Synapse1.0. There were only three resampling methods, ROS, AHC, and SPD, that could achieve the highest

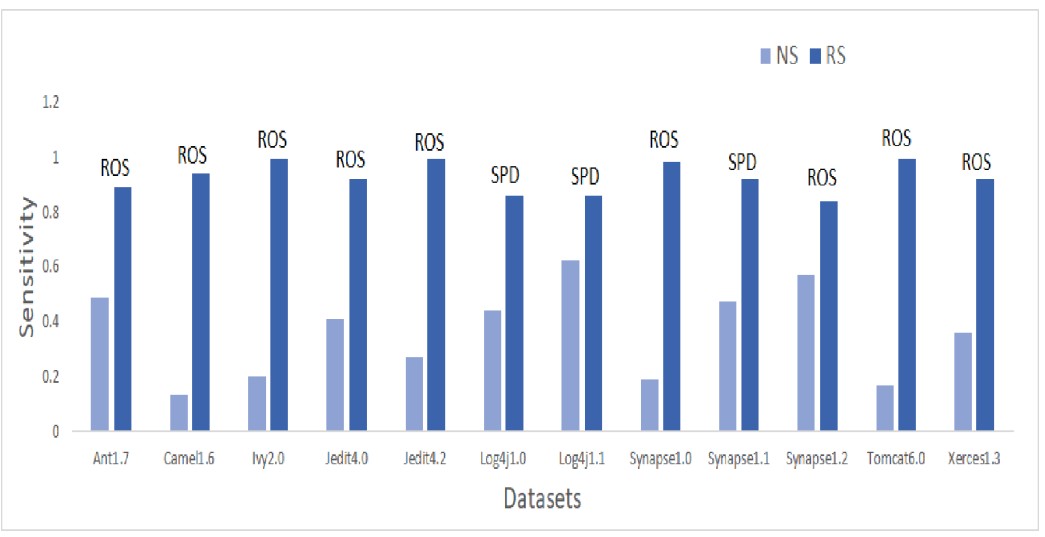

**Figure 7 Dataset-wise comparison of median sensitivity values of NS and RS.**

Balance value greater than 90. ROS was able to generate a Balance value of 95.58 for Ivy2.0. Comparatively, undersampling techniques, RUS and OSS, had a maximum Balance value of only 77.36 and 80.02 respectively. The highest median and mean values for all datasets except Synapse1.2 are attained by either SPD or ROS. Synapse 1.2 got the highest mean value with NCL and undersampling method. 60.9% of cases have a Balance greater than 70. Therefore, there was 357% of growth in median values of Balance when resampling is done as compared to NS.

*GMean analysis*

Similar patterns were also observed in GMean. Bar graphs in Fig. 6 portray the better predictive capabilities of ML techniques with ROS. The highest mean and median values are attained by ROS and SPD for all datasets except Synapse1.2. NCL gave the best results for mean and median values of Synapse1.2. 80.7% of cases have GMean greater than 0.65 for resampling methods. AHC, though not having any maximum values, but have consistent performance for all the datasets. With NS, only 3.9% of cases could achieve a GMean value greater than 0.75. With resampling methods, the number of cases for GMean values greater than or equal to 0.75 has elevated to 42.7% with the 1,182% of improvement. Comparing the oversampling and undersampling methods, oversampling methods thrived in predicting defects efficiently.

*Sensitivity analysis*

Sensitivity values are illustrated in the bar graph presented in Fig. 7. CNNTL worked best for Log4j1.1 and Synapse1.2 with an average Sensitivity value of 0.85 and 0.83 respectively. For other datasets, ROS outperformed other resampling methods. After applying resampling methods, Sensitivity increases above 0.9 for 11.9% of the developed models. 53% of resampling-based models have a Sensitivity greater than 0.7 as compared to only

**Table 7 Percentage improvement in performance measures after resampling methods are used.**

| Datasets | AUC growth | | Balance growth | | GMean growth | | Sensitivity growth | |
|---|---|---|---|---|---|---|---|---|
| | Max | Median | Max | Median | Max | Median | Max | Median |
| Ant1.7 | 18.1% | 1.3% | 30.9% | 16.4% | 27.8% | 12.1% | 67.2% | 44.9% |
| Camel1.6 | 34.2% | 4.4% | 69.8% | 61.3% | 68.5% | 72.2% | 180.0% | 353.8% |
| Ivy2.0 | 19.3% | 13.7% | 62.3% | 74.7% | 54.8% | 75.0% | 130.2% | 250.0% |
| Jedit4.0 | 24.1% | 1.3% | 34.9% | 23.6% | 32.9% | 17.7% | 75.0% | 73.2% |
| Jedit4.2 | 19.0% | 1.2% | 66.7% | 49.5% | 57.4% | 45.1% | 147.5% | 155.6% |
| Log4j1.0 | 14.1% | 9.2% | 34.9% | 26.0% | 31.0% | 20.6% | 75.0% | 68.2% |
| Log4j1.1 | 12.8% | 0.0% | 16.1% | 2.2% | 15.2% | 1.4% | 41.4% | 19.4% |
| Synapse1.0 | 33.8% | 23.9% | 38.7% | 86.8% | 33.8% | 86.0% | 76.8% | 326.3% |
| Synapse1.1 | 21.8% | 0.0% | 35.9% | 13.1% | 36.4% | 9.4% | 72.7% | 44.7% |
| Synapse1.2 | 14.6% | 3.9% | 21.5% | 7.2% | 21.1% | 5.9% | 47.6% | 33.3% |
| Tomcat6.0 | 23.5% | 3.8% | 70.8% | 77.7% | 62.7% | 85.0% | 160.5% | 288.2% |
| Xerces1.3 | 16.9% | 5.2% | 45.8% | 31.9% | 38.8% | 23.7% | 93.9% | 88.9% |
| Overall | 16.3% | 5.3% | 24.9% | 29.1% | 21.5% | 21.7% | 41.4% | 82.1% |

0.6% of cases in the NS scenario. This accounts for a whopping 9,440% of improvement in ML models having a Sensitivity value greater than 0.7.

Therefore, it can be concluded that the predictive capability of ML models has immensely improved after treating imbalanced data properly using resampling methods.

### Comparison of resampling-based ML models with NS models

A comparison of the results of models developed without employing resampling methods and models developed using resampling methods provides an answer to RQ3b. 1800 ML models were constructed for 12 datasets with ten resampling methods. The results were compared based on the maximum value attained and averaged median value achieved in each dataset. Percentage Improvement in Performance Measures after resampling methods are used is presented in Table 7. Analysis of Table 7 shows that there is a positive increment in all the values of maximum or median for the four performance evaluators when resampling methods are employed to build SDP models. This proves that there is a definite improvement in resampling-based ML models than the NS models when evaluated based on AUC, Balance, GMean, and Sensitivity.

For AUC, the overall percentage growth for the median value was 6.3%. The maximum AUC value achieved in the NS case is 0.86 for Log4j1.1 which increases to 0.97 when resampling methods are applied to it. Jedit4.2 and Tomcat6.0 were able to attain a maximum AUC value of 1 showing 18.2% and 22.2% of the increase. For Synapse1.0, on the application of resampling methods, the maximum value depicts the incremental growth of 33.8% and the respective median growth corresponds to 24.3%.

The increase in median values of Balance and GMean for all the datasets is 29.1% and 21.2% respectively with resampling methods. Synapse1.0, Tomcat6.0, Ivy2.0, and Camel1.6 has illustrated more than 60% of improvement in Balance median values and more than 70% improvement in GMean median values. The maximum Balance value

**Table 8 Friedman rankings for resampling methods with AUC, Balance, GMean, and Sensitivity.**

| | Rank 1 | Rank 2 | Rank 3 | Rank 4 | Rank 5 | Rank 6 | Rank 7 | Rank 8 | Rank 9 | Rank 10 | Rank 11 |
|---|---|---|---|---|---|---|---|---|---|---|---|
| AUC | ROS (10.11) | AHC (9.13) | SPD (8.58) | NCL (8.04) | SMT (7.06) | ADSYN (5.55) | NS (4.3) | SLSMT (4.05) | RUS (3.37) | CNNTL (3.08) | OSS (2.68) |
| Balance | ROS (10.08) | AHC (9.01) | SPD (7.93) | SMT (7.56) | NCL (6.6) | ADSYN (6.23) | SLSMT (5.09) | RUS (4.42) | CNNTL (3.85) | OSS (3.41) | NS (1.77) |
| GMean | ROS (10.15) | AHC (9.01) | SPD (7.91) | SMT (7.46) | NCL (6.95) | ADSYN (6.11) | SLSMT (4.88) | RUS (4.36) | CNNTL (3.66) | OSS (3.3) | NS (2.16) |
| Sensitivity | ROS (9.89) | AHC (8.83) | SMT (7.06) | SPD (7.06) | ADSYN (7.01) | CNNTL (6.48) | NCL (5.29) | SLSMT (5.2) | RUS (4.01) | OSS (3.9) | NS (1.22) |

Note:
  Best resampling technique is highlighted in bold.

gained by models with resampling methods is 95.58 for Ivy2.0 which was earlier 58.88. Similarly, the Sensitivity median values have shown a remarkable improvement of 84.6%. The median value of NS was 0.39 when all datasets were considered together. This value was raised to 0.71 on the application of resampling methods.

Answer to RQ3b—The results verify that there is an improvement in the performance of SDP models developed using ML techniques on the application of resampling methods.

### RQ4: Which resampling method outperforms the addressed undersampling and oversampling techniques for building an efficient SDP model?

This RQ addresses the effectiveness of 10 resampling methods that are investigated in this study for constructing good SDP models. For the experiments conducted, is there any particular resampling method that can be considered the best? In this direction, we conducted Friedman tests on performance evaluators to provide rankings to models built using resampling methods. The case when no resampling method was used is also included.

The Friedman test was used to find the difference between techniques statistically. Four hypotheses are formed for four different performance evaluators. The hypothesis formed to achieve this objective is stated as:

$H_i0$ (Null Hypothesis): There is no significant statistical difference between the performance of any of the defect prediction models developed after using resampling methods and models developed using original data, in terms of $PM_j$.

$H_ia$ (Alternate Hypothesis): There is a significant statistical difference between the performance of any of the defect prediction models developed after using resampling methods and model developed using original data, in terms of $PM_j$.

where i = 1 to 4 denoting $H_1$, $H_2$, $H_3$ and $H_4$ hypothesis and j = 1 to 4. $PM_1$ = AUC, $PM_2$ = Balance, $PM_3$ = GMean, and $PM_4$ = Sensitivity.

Table 8 provides the desired ranking of SDP models developed in this study for AUC, Balance, GMean, and Sensitivity. Best resampling technique is highlighted in bold for each performance measure in Table 8. The mean ranks of each resampling method and NS are shown in parentheses. As discussed, NS represents the scenario when no sampling technique is used, so, it represents the performance with original data. We evaluated the hypothesis at the 0.05 level of significance, i.e., 95% of the confidence interval. Rank 1 is the best rank and rank 11 is the worst rank. The *p*-values achieved for all performance evaluators are 0.000 and recorded in Table 8. As *p*-values were less than 0.05, we rejected

**Table 9 Pairwise comparison of resampling methods using Nemenyi test for AUC, Balance, GMean, and Sensitivity.**

| Pair | AUC | Balance | GMean | Sensitivity |
|------|-----|---------|-------|-------------|
| ROS-AHC | NS | NS | NS | NS |
| ROS-SPD | S+ | S+ | S+ | S+ |
| ROS-SMT | S+ | S+ | S+ | S+ |
| ROS-NCL | S+ | S+ | S+ | S+ |
| ROS-ADSYN | S+ | S+ | S+ | S+ |
| ROS-SLSMT | S+ | S+ | S+ | S+ |
| ROS-RUS | S+ | S+ | S+ | S+ |
| ROS-CNNTL | S+ | S+ | S+ | S+ |
| ROS-OSS | S+ | S+ | S+ | S+ |
| ROS-NS | S+ | S+ | S+ | S+ |

the null hypothesis and declared that there was a significant difference between resampling methods applied to developed SDP models.

Table 8 shows that ROS and AHC have unanimously scored Rank 1 and Rank 2 respectively for all the performance evaluators. Out of four undersampling methods, only one, i.e., NCL can make its space in the first seven positions for the reliable performance evaluators—AUC, Balance, and GMean. OSS, CNNTL, and RUS have acquired positions in the last four ranks with AUC, Balance, and GMean. These rankings clearly state the supremacy of oversampling methods over undersampling methods.

For Balance, GMean, and Sensitivity, NS case is ranked last. Therefore, results statistically approved the visualization in RQ3 that usage of resampling methods improved the predictive power of SDP models.

We have used Kendall's coefficient of concordance to assess the effect size. It is a quantitative measure of the magnitude of the experimental effect. Kendall in Table 8 shows the value for Kendall's coefficient of concordance. Its value ranges from 0 to 1. It reflects the degree of agreement. The Kendall value is 0.656, 0.589, 0.587, and 0.538 for AUC, Balance, GMean, and Sensitivity respectively. As the values are greater than 0.5 but less than 0.7, therefore, its effect is moderate.

The Friedman test shows whether there is an overall difference or not in the model performances, but if there is a difference, it fails to further identify the pairwise difference, i.e., exactly which technique is significantly different from other. For this, a post-hoc analysis was conducted using Nemenyi Test on overall datasets and the comparative pairwise performance of all the resampling methods was evaluated with ROS. The Nemenyi test was carried out at the $\alpha = 0.05$ level of significance. Table 9 summarizes the Nemenyi test results for AUC, GMean, Balance, and Sensitivity. 'S+' represents 'significantly better' results. If the difference between the mean ranks of the two techniques is less than the value of critical distance (CD), there is no significant difference in the 95% confidence interval. If the difference is greater than the CD value, the technique with a higher rank is considered statistically better. The computed CD value for the Nemenyi test conducted for resampling methods was 1.135. It can be inferred from Table 9 that ROS

**Table 10 Friedman rankings for ML techniques with AUC, balance, gmean, and sensitivity.**

|  | AUC | Balance | GMean | Sensitivity |
|---|---|---|---|---|
| Rank 1 | **Kstar (14.41)** | **ABM1 (13.41)** | **ABM1 (13.5)** | **RT (12.62)** |
| Rank 2 | ABM1 (13.75) | IBk (12.66) | IBk (12.91) | IBk (12.54) |
| Rank 3 | BAG (13.37) | RT (12.16) | RT (12.58) | Kstar (12.33) |
| Rank 4 | RSS (12.29) | BAG (12.12) | BAG (11.75) | ABM1 (12.2) |
| Rank 5 | IBk (9.83) | Kstar (11.08) | Kstar (11.25) | LMT (10.95) |
| Rank 6 | RT (9.2) | RSS (10.33) | RSS (9.58) | PART (9.95) |
| Rank 7 | LMT (8.58) | PART (9.04) | PART (9.04) | J48 (9.83) |
| Rank 8 | J48 (7.62) | LMT (9) | LMT (9) | BAG (9.54) |
| Rank 9 | PART (7.58) | J48 (8.58) | J48 (8.7) | RSS (8.75) |
| Rank 10 | LB (6.08) | MLP (5.45) | MLP (5.75) | LB (5.25) |
| Rank 11 | ICO (5.54) | ICO (4.87) | ICO (4.87) | MLP (4.91) |
| Rank 12 | MLP (4.75) | LB (4.83) | LB (4.7) | ICO (4.83) |
| Rank 13 | LR (2.79) | SL (2.75) | SL (2.66) | SL (2.66) |
| Rank 14 | SL (2.5) | LR (2.41) | LR (2.33) | LR (2.41) |
| Rank 15 | NB (1.66) | NB (1.25) | NB (1.33) | NB (1.16) |
| $p$-value | **0.000** | **0.000** | **0.000** | **0.000** |
| Kendall | 0.876 | 0.835 | 0.844 | 0.838 |

**Note:**
First ranked ML techniques and significant $p$-values attained for different performance measures are indicated in bold.

has comparable performance with AHC and exhibits statistically better performance than all other compared scenarios based on AUC, Balance, GMean, and Sensitivity.

Answer to RQ4: Oversampling methods resulted in better SDP models than undersampling methods. ROS and AHC emerged as the statistically better resampling method in terms of AUC, Balance, GMean, and Sensitivity.

## RQ5: which ML technique performs the best for SDP based on resampled data?

To answer this RQ, we performed the Friedman test for 15 ML techniques by considering ROS-based models. The Friedman rankings are recorded in Table 10 with Rank 1 as the best rank. The ML Technique that achieved best rank is bold in Table 10. The Friedman test was held at the 0.05 significance level with a degree of freedom of 14. The null hypothesis is set as there is no difference between comparative performances of ROS-based models for different ML techniques. In Table 10, significant $p$-values ($p$-values < 0.05) are highlighted in bold. The $p$-value for each performance evaluator was 0.000. Therefore, the results are considered 95% significant. We refute the null hypothesis as there is a significant difference amongst performances of models built using different ML techniques. The Kendall value for AUC, Balance, GMean, and Sensitivity for ML techniques with ROS-based models is 0.876, 0.835, 0.844, and 0.838. All the four performance evaluators have a Kendall value greater than or equal to 0.835. This signifies that rankings for different datasets are approximate 83.5% similar and hence, increases the

**Table 11 Pairwise comparison of ML techniques using Nemenyi Test for AUC, Balance, GMean, and Sensitivity.**

| Pair | AUC | Balance | GMean | Sensitivity |
|------|-----|---------|-------|-------------|
| ABM1-IBk | NS | NS | NS | NS |
| ABM1-RT | NS | NS | NS | NS |
| ABM1-BAG | NS | NS | NS | NS |
| ABM1-Kstar | NS | NS | NS | NS |
| ABM1-RSS | NS | NS | NS | NS |
| ABM1-PART | NS | NS | NS | NS |
| ABM1-LMT | NS | NS | NS | NS |
| ABM1-J48 | NS | NS | NS | NS |
| ABM1-MLP | **S+** | **S+** | **S+** | **S+** |
| ABM1-ICO | **S+** | **S+** | **S+** | **S+** |
| ABM1-LB | **S+** | **S+** | **S+** | **S+** |
| ABM1-SL | **S+** | **S+** | **S+** | **S+** |
| ABM1-LR | **S+** | **S+** | **S+** | **S+** |
| ABM1-NB | **S+** | **S+** | **S+** | **S+** |

**Note:**
Significant results are in bold.

reliability and credibility of the Friedman statistical results. The impact of differences in results is high.

This can be observed from Table 10 that Kstar, IBk, ABM1, RT, and RSS techniques incited better prediction models than other ML techniques.

Kstar and IBk are the variants of the nearest neighbor techniques. These ML techniques provide good results when there is a large number of samples irrespective of the data distribution. Nearest neighbors are instance-based fast learners. Mean ranks are inscribed in brackets. The mean rank of KStar is 14.41 for AUC which is the highest in the pool. The rank of IBk is second when the models are evaluated based on Balance, GMean, and Sensitivity. The mean rank of IBk is 12.66, 12.91, and 12.33 for Balance, GMean, and Sensitivity respectively.

ABM1, RT, Bag, and RSS have also attained high ranks. ABM1 is the first ranker in Balance and GMean with a mean rank of 13.41 and 13.5 respectively. ABM1 got 2nd rank with AUC and 4th rank with sensitivity. For the stable performance evaluators, ABM1, Bag, RT, and RSS appeared in the first six ranks, proving their competency in the ML world. These four techniques are ensembles that are considered robust in dealing with imbalanced data. The crux of ensemble techniques is to cover the weakness of the base ML technique and combine them to reduce bias or variance and enhance its predictive capability.

The last three ranks for all the performance evaluators are grabbed by the statistical learners: Naïve Bayes, Simple Logistic, and Logistic Regression. These ML techniques, in contrast, were able to generate good prediction models when no resampling method was involved in model construction. The balancing of both the classes boosted the performance of other classifiers, especially ensembles and nearest neighbors, and resulted in their unacceptable performance.

The post-hoc analysis is carried out using the Nemenyi Test and results are computed with CD = 6.191. Pairwise Comparison of ML Techniques using Nemenyi Test for AUC, Balance, GMean, and Sensitivity are presented in Table 11 and significant results are in bold. ABM1 was paired with other ML techniques and it was found comparable with IBk, Kstar, RT, BAG, RSS, PART, LMT, and J48. ABM1 was found statistically significant than MLP, ICO, LB, SL, LR, and NB.

Answer to RQ5: Ensembles and nearest neighbors performed the best for SDP in imbalanced data with a random oversampling method.

## DISCUSSIONS

Feature Selection is an important activity in software quality predictive modeling. The most widely accepted feature selection technique in literature is CFS. Through this study, we provided the important subset of features to the software practitioners. We identified that coupling and cohesion are the most important internal quality attributes that developers should focus on while building software to avoid defects. Defects in software were found to be highly correlated to LCOM, RFC, Ca, and Ce.

Median values of Sensitivity have improved by 45.7%, 341.9%, 250%, 72.2%, 154.8%, 66.7%, 18.9%, 330%, 46.5%, 32.8%, 293.3%, and 88.8% for Ant1.7, Camel1.6, Ivy2.0, Jedit4.0, Jedit4.2, Log4j1.0, Log4j1.1, Synapse1.0, Synapse1.1, Synapse1.2, Tomcat6.0, and Xerces1.3 respectively after applying resampling methods.

Median values of Balance have improved by 16.4%, 61.3%, 74.7%, 23.6%, 49.5%, 26%, 2.2%, 86.8%, 13.1%, 7.2%, 77.7%, and 31.9% for Ant1.7, Camel1.6, Ivy2.0, Jedit4.0, Jedit4.2, Log4j1.0, Log4j1.1, Synapse1.0, Synapse1.1, Synapse1.2, Tomcat6.0, and Xerces1.3 respectively after applying resampling methods.

Median values of G-Mean have improved by 13.1%, 73.7%, 73.5%, 16.9%, 44.7%, 20.1%, 0.4%, 88.3%, 9%, 5.5%, 83.5%, and 24.2% for Ant1.7, Camel1.6, Ivy2.0, Jedit4.0, Jedit4.2, Log4j1.0, Log4j1.1, Synapse1.0, Synapse1.1, Synapse1.2, Tomcat6.0, and Xerces1.3 respectively after applying resampling methods.

Median values of ROC-AUC have improved by 1.9%, 3.8%, 14.3.7%, 1.9%, 1.3%, 8.7%, 0.6%, 24.3%, 0.7%, 3.3%, 4.7%, and 5.8% for Ant1.7, Camel1.6, Ivy2.0, Jedit4.0, Jedit4.2, Log4j1.0, Log4j1.1, Synapse1.0, Synapse1.1, Synapse1.2, Tomcat6.0, and Xerces1.3 respectively after applying resampling methods.

For the datasets considered, ROS and AHC were significantly better than other resampling methods. Better model predictions can be achieved by incorporating oversampling methods than the undersampling methods. Our findings are in the agreement with the *Wang & Yao (2013)* conclusions. They proved that resampling methods improve defect prediction and, in their settings, the AdaBoost ensemble gave the best performance.

Resampling methods did not improve the defect predictive capability of statistical techniques. Developers and researchers should prefer ensemble methods for software quality predictive modeling.

# VALIDITY THREATS

## Conclusion validity

Conclusion validity threat is a threat to statistical validity and this indicates that results of the empirical study are not properly analyzed and validated. To avoid this threat, ten-fold cross-validation is performed. The Friedman test and Nemenyi test during post hoc analysis strengthens conclusion validity further. In this study, both the statistical validation techniques are nom parametric in nature. Non-parametric tests are not based on any assumptions for underlying data and, therefore, applicable to the selected datasets. This reinforces the analysis of the relationship between independent variables and dependent variables and hence enhancing conclusion validity.

## Internal validity

Applying resampling methods results in the change in the original ratio of defective and non-defective classes. This would be affecting the causal relationship between independent and dependent variables resulting in internal validity bias in our study. However, we have used stable performance evaluators, GMean and AUC, to assess the performance of different SDP that help to reduce this threat. We have also used Sensitivity that takes care of the proper classification of defective classes, which is one of the major requirements of our problem domain. So, judicious selection of performance evaluators may have reduced some effect of internal validation threat.

## Construct validity

Construct validity ensures the correctness of the way of measuring the independent and dependent variables of the study. It also emphasizes whether the variables are correctly mapped to the concept that they are representing. The Independent variables of this study are object-oriented software metrics and the dependent variable of this study is 'Defect' representing the absence or presence of a defect in the software module. Chosen independent variables and dependent variables are widely used in defect prediction studies and thus builds confidence in the removal of construct validity threats from our study. Hence, any construct validity threats may not exist in our study.

## External validity

External validity refers to the extent to which the results of the study are widely applicable. Whether the conclusions of the concerned study can be generalized or not? The datasets used in this study are available publicly. Concerned software are written in the JAVA language. Thus, results validity holds for similar situations only. Results may not be valid for proprietary software. Resampling methods are implemented with default parameter settings in KEEL (*Alcalá-Fdez et al., 2011*), and ML parameter settings are provided. Therefore, this study can be reproduced without any complications. This minimizes the external validity threat in this study.

## CONCLUSIONS

This study evaluates the effect of resampling methods on various machine learning models for defect prediction using Apache software. In total, 1,980 models were built and the performances of models were empirically compared using stable performance evaluators such as GMean, Balance, and AUC. Important features were selected using CFS and ten-fold cross-validation was executed while training the model. Six oversampling and four undersampling methods were analyzed with 15 ML techniques and results were statistically validated by using the Friedman test followed by Nemenyi *post hoc* analysis. The use of statistical tests reinforces the correctness of results. CFS is incorporated to minimize the multicollinearity effect whereas resampling methods reduces the model bias and assures that predictions are not affected by majority class label. This study reinsures that SDP models developed with resampling methods enhance their predictive capability as compared to models developed without resampling methods. The results of the Friedman test strongly advocate the use of the random oversampling method for the improved predictive capability of SDP classifiers for imbalanced data. Apart from ROS, AHC and SMT also demonstrated good predictive capability for uncovering defects. The Nemenyi test eradicates family-wise error and concluded ROS and AHC to be significantly and statistically better than other resampling methods. Models developed using oversampling methods illustrated better defect prediction capability than that of undersampling methods. Handling CIP using ROS and AHC will aid developers and software practitioners in detecting defects effectively in the early stages of software development reducing testing cost and effort. Resampling methods greatly improved the performance of ensemble methods.

Future direction involves the inclusion of more imbalanced datasets and investigates the impact of resampling methods on them. Datasets can be taken of any prevalent language different than JAVA like C#. There is a dire need for a benchmark study that compares all the existing resampling solutions in the literature with the common experimental settings. Optimized versions of base resampling methods like RUS or ROS can be proposed. Further, we will also like to explore the consequences of resampling methods with search-based techniques for the classification of software defects. Instead of predicting defects, a framework can besides be utilized to envisage defect severity.

### Funding
The authors received no funding for this work.

### Competing Interests
The authors declare that they have no competing interests.

### Author Contributions
- Ruchika Malhotra conceived and designed the experiments, analyzed the data, performed the computation work, authored or reviewed drafts of the paper, supervision, and approved the final draft.

- Juhi Jain conceived and designed the experiments, performed the experiments, analyzed the data, performed the computation work, prepared figures and/or tables, authored or reviewed drafts of the paper, and approved the final draft.

## Data Availability

Raw data are available as Supplemental Files 1 to 12. These files represent the datasets used in this study.

Regarding code availability: We used KEEL and WEKA software for implementing resampling methods and ML techniques. Both these tools are opensource tools.

## Supplemental Information

Supplemental information for this article can be found online at http://dx.doi.org/10.7717/peerj-cs.573#supplemental-information.

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
