# Peer review of "Predicting defects in imbalanced data using resampling methods: an empirical investigation"

_PeerJ Computer Science, doi:10.7717/peerj-cs.573_

## Round 0.1 · original submission · Major Revisions

The reviewers and I agree that this is an interesting article, but it needs a major revision to be further considered for publication. My summary of the major issues is:

1. Differentiation from existing similar studies
2. Description and completeness of the statistical analysis

But please take all issues mentioned in the reviews into account.

Reviewer 1 ·

Basic reporting

The structure or the organization of the paper is not very clear to me. I understand that the authors of the paper are trying to investigate the impact of class reblancers on defect prediction models. However, I am not sure why the authors investigate the features that are selected consistently by a CFS method. They do not use this finding anywhere else. Furthermore, they assert that using all the features might lead to overfitting of the models. However, the studied defect datasets do not have more than 30 features! So I would recommend that the authors explain why they study a feature selection method along with class rebalancing if there is no connection between the findings.


I recommend the authors to replace table 6, 7, 8,9 ,10, 11 with side-by-side boxplots for better readability

Experimental design

several prior studies (e.g., https://arxiv.org/abs/1609.01759) find that tuning the machine learning classifiers is extremely important for the defect prediction models to perform well. In this case, it could particularly be an important confounder that affects the results. I would recommend the authors to tune the models that they use in their study

I am not clear about how the statistical tests are used. For instance, the authors say they used a Friedman test followed by a Wilcoxon test to rank the performance of the classifiers. However, a friedman test to my knowledge doesn’t generate ranking and a Wilcoxon test is only a parwise comparison. So I am not sure how the authors arrive at the ranking. I would urge the authors to explain the statistical tests that they used to arrive at their results a little clearly

Similarly, to rank the results of Friedman test, a post-hoc nemenyi test is the typical suggestion, rather than the Wilcoxon. Hence I would suggest the authors to use that (e.g., you can follow this paper: https://arxiv.org/abs/1707.09281, https://joss.theoj.org/papers/10.21105/joss.02173)

If Wilcoxon is used to compare multiple pairs, you need to use a p-value correction like a Bonferroni correction for the results to be reliable.

Validity of the findings

Major concern: Some of the key related works are missing. For instance, a recent TSE (https://arxiv.org/abs/1801.10269) and an ICSE (https://arxiv.org/pdf/1705.03697.pdf) study through a comprehensive study finds that SMOTE is indeed better than other class rebalancing methods. These studies are not discussed. I would recommend the authors to consider these studies and position their findings in the context of these studies.

More along my previous point, the aforementioned papers assert that tuning the SMOTE is important for the benefits of SMOTE to shine through. Tantithamthavorn et. al. (https://arxiv.org/abs/1801.10269) in particular find that tuned SMOTE helps defect prediction models perform better than unbalanced classifier. They also compare several class rebalancing methods (though not as comprehensive). So I would like the authors to position their findings in the context of this paper. Therefore, I would suggest the authors to include the datasets used by Tantithamthavorn et. al. also in their studies to see if their findings agree with Tantithamthavorn et. al. or if they could refute them.

Because the statistical issues outlined earlier I am not sure if the results are reliable.

Additional comments

1. The paper compares an impressive array of class rebalancing methods and defection prediction classifiers.
2. The paper comprehensively tries to estimate the impact of the class rebalancing method by experimenting with multiple configurations.

Reviewer 2 ·

Basic reporting

- In general, the manuscript is clear and well-written, however it lacks a Discussion section which would be useful to present a general overview of the findings and discuss the results on a higher level.

- The study can also benefit from a small description of the research questions under study to make it more clear to the reader.

- While Figure 1 might be comprehensive, it would be better to include a small description which might make it easier for the reader to understand the full experimental process of the study.

- The paper contains grammatical/spelling errors that can be fixed with a detailed proofread, some of them are listed below:
Line 25: “…softwares…” --> “…software…”
Line 37: “…with uncovering the probable…” --> “…with uncovering probable…”
Line 44: “…becomes a difficult task.” --> “…become difficult tasks.”
Line 52: “…trained on the similar…” --> “…trained on similar…”
Line 65: “…it is widely…” --> “…it is a widely…”
Line 66: “…and emerged…” --> “…and it has emerged…”
Line 114: “…for existing ones.” --> “…from existing ones.”
Line 303: “…by Wilcoxon signed-rank test…” --> “…by the Wilcoxon signed-rank test…”
Line 341: “…IQA is scribed…” --> “…IQA is described…”
Line 350: “…66.67%%...” --> “…66.67%...”

Experimental design

The paper includes a good experimental design of a large empirical study by including a set of widely used ML techniques and datasets, a set of non-biased evaluation measures and statistical tests. However, statistical significance cannot be assessed alone without analysing the effect size (Arcuri and Briand, 2004 https://dl.acm.org/doi/10.1002/stvr.1486). The effect size is a quantitative measure of the magnitude of the experimental effect and it would give the reader a better understanding of the impact of the difference in results.

Some of the datasets used in this study describe systems that consist of multiple versions, for example Log4j1.0 and Log4j1.1. Is there any reason 10-fold cross validation was favoured and applied over the more realistic cross-version defect prediction scenario whereby the model is trained on an older version and tested on the most recent one?

Validity of the findings

The study includes a well-developed related work section but fails to position the work with respect to existing ones, for example, the work of Wang and Yao, 2013. This undermines the contribution and novelty of the study presented which should be better highlighted in the manuscript with respect to the large number of existing studies tackling the data imbalance problem and comparing over- and under-sampling techniques.

---

## Round 0.2 · Minor Revisions

The reviewer is mostly happy with the revision and so am I. I suggest including their recommendation on:

a. Rework the related work section to better highlight the differences to related work.

b. Move the summary paragraph from section 5 to section 2.

Reviewer 2 ·

Basic reporting

The authors have addressed the previously raised comments

Experimental design

The authors have addressed the previously raised comments

Validity of the findings

The authors have mainly addressed the previously raised comments, thanks. I only still have a concern regarding the position of the paper with respect to related work. I believe
it would be easier for the reader to understand the contribution of the work if the differences between the proposed study and previous work are highlighted in a better and clearer manner. I also believe that the newly added paragraph about previous similar work would be a better fit in Section 2 (Related Work) rather than in Section 5 (Discussion).

Additional comments

no comment

---

## Round 0.3 · accepted · Accept

Thanks for incorporating the suggestions from the last round of reviews. They look fine to me! I think the article is now ready to be published.